# Current Advances in Cellular Approaches for Pathophysiology and Treatment of Polycystic Ovary Syndrome

**DOI:** 10.3390/cells12172189

**Published:** 2023-08-31

**Authors:** Yi-Ru Tsai, Yen-Nung Liao, Hong-Yo Kang

**Affiliations:** 1Graduate Institute of Clinical Medical Sciences, College of Medicine, Chang Gung University, Taoyuan City 333, Taiwan; 2An-Ten Obstetrics and Gynecology Clinic, Kaohsiung City 802, Taiwan; 3Department of Chinese Medicine, Kaohsiung Chang Gung Memorial Hospital, College of Medicine, Chang Gung University, Kaohsiung City 833, Taiwan; 4Department of Biological Science, National Sun Yat-sen University, Kaohsiung 804, Taiwan; 5Center for Hormone and Reproductive Medicine Research, Department of Obstetrics and Gynecology, Kaohsiung Chang Gung Memorial Hospital, College of Medicine, Chang Gung University, Kaohsiung City 833, Taiwan; 6Division of Endocrinology and Metabolism, Department of Internal Medicine, Kaohsiung Chang Gung Memorial Hospital, College of Medicine, Chang Gung University, Kaohsiung City 833, Taiwan

**Keywords:** polycystic ovary syndrome, hyperandrogenism, androgen receptor, granulosa cells, folliculogenesis

## Abstract

Polycystic ovary syndrome (PCOS) is a prevalent gynecological and endocrine disorder that results in irregular menstruation, incomplete follicular development, disrupted ovulation, and reduced fertility rates among affected women of reproductive age. While these symptoms can be managed through appropriate medication and lifestyle interventions, both etiology and treatment options remain limited. Here we provide a comprehensive overview of the latest advancements in cellular approaches utilized for investigating the pathophysiology of PCOS through in vitro cell models, to avoid the confounding systemic effects such as in vitro fertilization (IVF) therapy. The primary objective is to enhance the understanding of abnormalities in PCOS-associated folliculogenesis, particularly focusing on the aberrant roles of granulosa cells and other relevant cell types. Furthermore, this article encompasses analyses of the mechanisms and signaling pathways, microRNA expression and target genes altered in PCOS, and explores the pharmacological approaches considered as potential treatments. By summarizing the aforementioned key findings, this article not only allows us to appreciate the value of using in vitro cell models, but also provides guidance for selecting suitable research models to facilitate the identification of potential treatments and understand the pathophysiology of PCOS at the cellular level.

## 1. Introduction

Polycystic ovarian syndrome (PCOS) is a common heterogeneous endocrine disorder that affects 7–12% of reproductive-age women and is significantly associated with infertility. The primary symptoms of PCOS include hyperandrogenism, absent ovulation, and polycystic ovaries. Currently, there are no definitive diagnostic criteria for PCOS, as its pathology involves multiple factors such as endocrinology and gynecology, making it difficult to categorize [1,2,3,4]. PCOS is characterized by an increased density of small pre-antral follicles and a higher proportion of early growing follicles, accompanied by abnormal granulosa cell proliferation. Additionally, PCOS is associated with the apoptosis of granulosa cells in antral follicles [5,6,7]. Granulosa cells (GCs) form a cuboidal layer on the surface of the oocyte and secrete steroid hormones that play a crucial role in follicle development, making them a key focus in PCOS pathogenesis [5,8]. During oocyte development, the oocyte and its surrounding granulosa cells exhibit a mutual interdependence, crucial for providing the oocyte with essential nutrients and growth regulators. In return, the oocyte plays a role in promoting the growth and differentiation of the granulosa cells. There is evidence to indicate that dysfunction in these cellular interactions may play a role in the abnormal folliculogenesis observed in PCOS [9]. Because the molecular mechanism responsible for triggering PCOS is unknown, it is difficult to offer guidance to women and healthcare professionals regarding the condition. Hence, discovering the specific mechanism that causes PCOS has become an area of intensive research.

In recent years, several in vitro models, including cell lines and animal models, have been developed to study PCOS due to ethical constraints in human studies. However, despite conducting numerous experiments, the underlying pathophysiology of PCOS still remains unclear. Nonetheless, the development of these experimental models holds significant potential in advancing our understanding of the condition and identifying the most effective treatment options [4]. Herein, we focus on up-to-date in vitro cell models for PCOS and summarize the mechanisms proposed or discussed in PCOS-like phenotypes in these models. We also briefly summarize the related mechanisms in hormonal and genetic alterations in developed PCOS, as well as the common pathophysiology that accompanies metabolic syndrome. We also summarize the common cellular/molecular mechanisms of drugs, compounds, and traditional Chinese herbal medicines, along with miRNA expression and specific genes implicated in PCOS pathogenesis. Finally, we discuss the effects of cells other than ovarian tissue on PCOS.

## 2. In Vitro Cell Models for PCOS research

Established animal models with PCOS characteristic are important for studying the pathophysiology and etiology of the disorder. Moreover, animal-derived cells or cell lines with an indefinite lifespan offer cost-effective and time-efficient alternatives for screening a broad spectrum of drugs before progressing to in-vivo models. Consequently, they provide a reliable and consistent cell source for experimentation. Figure 1 summarized the common in vitro cell models for PCOS research, as we discussed below.

### 2.1. Human Ovarian Granulosa Cell Lines

In vitro human cell line models have been used in fundamental research, including studies on PCOS. As outlined in Table 1, numerous human GC lines have been established successfully, including long-term cultures from human tumor cells [10,11,12,13,14,15] and immortalization through oncoviruses [16,17,18,19], lentiviruses [20,21,22,23], or retroviral systems [24,25]. GC function is defined by three distinct properties: (1) the production of 17β-estradiol in response to follicle stimulating hormone (FSH) through the action of P450 aromatase [26]; (2) support for oocyte growth during the development of the surrounding follicle [27,28]; and (3) the presence of specific molecular markers of apoptosis involved in follicle atresia [29]. To confirm the functional characteristics of these immortalized GCs, various measurement parameters were investigated for validation. GCs are essential in the ovary, playing a significant role in folliculogenesis and oocyte maturation. One of their important functions is secretion of steroid hormones, such as estrogen, progesterone, and AMH. Moreover, FSH receptor (FSHR) and aromatase are crucial markers of GCs. FSHR acts as an upstream regulator of aromatase, which plays a pivotal role in the production of estrogen by GCs through the aromatization of androstenedione. The level of sex steroid hormones in the culture medium can be measured to determine aromatase activity [24].

Most of the human GC cell lines described in Table 1 exhibit aromatase activity, although they are not all FSH-dependent. Additionally, several GC lines, including SVOG, HGL5, and HO23, are typically isolated and established by transfection of luteinized GCs obtained from IVF programs [16,17,18,19,22,23]. Due to the administration of human chorionic gonadotropin (hCG), luteinized GCs do not express a functional FSHR [30]. Both GCN and GCP, which are established from normal and PCOS GCs, express specific granulosa markers such as FSHR and aromatase, and their expressions of these markers are similar to those of primary GCs. These two GC lines provide a good in vitro model for understanding the cellular mechanisms of the normal and PCOS human ovary [24,25]. KGN, COV434, and HGrC1 were useful models because they maintain most physiological activities, such as the expression of functional FSHR and an FSH-dependent increase in aromatase activity [10,11,12,13,14,20,21]. Among these cell lines, KGN is particularly advantageous as it retains the physiologic features of GCs and exhibits a stable, long-term proliferative capability. Moreover, as it does not produce endogenous steroids and responds well to gonadotropins, KGN serves as a suitable model for the two-cell, two-gonadotropin model of ovarian steroidogenesis. Consequently, it is extensively used in research on the cellular functions of GCs and the molecular regulatory mechanisms involved in PCOS [10,19].
cells-12-02189-t001_Table 1Table 1Characteristics of human ovarian granulosa cell lines for PCOS research.Cell LineGCPGCNKGNCOV434HTOGHGrC1SVOGHGL5HO23**Population**IranianIranian.JapaneseNDJapaneseNDNDNDIsraeli**Age**323663277435NDAdultAdult**Category**Telomerase immortalized cell lineTelomerase immortalized cell lineCancer cell lineCancer cell lineCancer cell lineTransformed cell lineTransformed cell lineTransformed cell lineTransformed cell line**Disease**PCOSNormal patient with IVFOvarian granulosa cell tumorOvarian granulosa cell tumorOvarian granulosa cell tumorNDNDNDNormal patient with IVF**Transformed**hTERT/c-MychTERT/c-MycNDNDNDE6/E7SV40E6/E7SV40**Doubling time**75 h86 h46.4 h24 h25 h40 hND96 hND**Luteinized**✓✓✗✓ND✗✓✓✓**FSH responsive**✓✓✓✗ND✓✗✗✓**Androgen****responsive**NDND✓NDNDNDND✓✓**Aromatase**✓✓✓✓✓✓ND✓ND**P4**✓✓✓✓✓✓✓✓✓**E2**✓✓✓✓✓✓✗✓✗**AMH**✓✓✓✓NDNDNDND✓**FOXL2 gene**NDNDC134WWildtypeNDC134WNDNDND**Ref.**[24,25][24,25][10,11,12,13,31,32][11,14,31,32][15][20,21][16][22,23,32][17,18,19,32]ND, not determined; ✓, the feature shows in this cell line; ✗, the feature does not show in this cell line.


Both KGN and HGrC1 cell lines harbor a missense mutation in the Forkhead transcription factor gene, FOXL2 gene, which play a crucial role in the differentiation and function of the ovaries [33]. FOXL2 gene is involved in the proliferation and differentiation of GCs. Hence, there may be concerns regarding the use of KGN and HGrC1 in investigating the proliferation and apoptosis of GCs during normal folliculogenesis. By contrast, COV434, another granulosa cell tumor-derived cell line, has a wildtype FOXL2 genotype and does not express FOXL2 from a juvenile GC tumor, making it a better model for investigating these aspects [11,14,21]. 

Establishing appropriate in vitro GC cell line models can help in investigating the role of steroidogenesis, oogenesis, folliculogenesis, atresia, and luteinization in PCOS [19]. Many studies have used primary culture systems of GCs obtained from follicular fluids during oocyte retrieval in IVF programs. However, some limitations still exist in GC culturing such as low cell yield, difficulties in maintaining cell viability for extended cell generations and preparing uniform cell populations in sufficient amounts. By contrast, human GC cell lines have no limitations in cell number and show long-term viability and decreased proliferation.

### 2.2. Primary Ovary and Other Relevant Cell Lines for PCOS In Vitro Studies

Follicular fluid from patients undergoing IVF procedures is a common source of human GCs. Therefore, successfully isolating and purifying abundant levels of high-quality GCs from the follicular fluid is crucial for PCOS research [34,35]. As shown in Table 2, several human GC isolation techniques have been described in the literature, enabling us to investigate the development mechanisms of PCOS, including proliferation, apoptosis, insulin resistance (IR), and oxidative stress (OS) [36,37,38,39,40,41,42,43,44,45,46,47,48,49]. Hyperandrogenism is also a prominent feature of PCOS, characterized by excessive production of Δ4 steroids such as androstenedione (A4) and testosterone, which can lead to anovulation, oligomenorrhea, and infertility [50]. The increased expression of steroidogenic enzymes P450c17 (17α-hydroxylase, 17,20-lyase) and 3β-HSDII (3beta-hydroxysteroid dehydrogenase), which are essential for androgen production, causes enhanced androgen biosynthesis in PCOS [51,52,53]. NCI-H295R is a human adrenocortical cell line widely recognized as a valuable model for studying steroidogenesis [54,55,56,57,58,59]. These cells express all the genes that encode the steroidogenic enzymes found in all three layers of the adult adrenal cortex, including 3β-HSDII and P450c17.

#### 2.2.1. Primary GC Isolation from Medicine-Induced PCOS Mouse Models In Vivo

Using human models for PCOS research does have limitations due to difficulties in obtaining samples, as well as ethical or logistical reasons. Therefore, based on the high degree of evolutionary conservation of the mammalian reproductive system, the development of animal models is essential for exploring the pathophysiology of PCOS both in vivo and in vitro. As shown in Table 2, PCOS models in animals can be induced using compounds such as dehydroepiandrosterone (DHEA), dihydrotestosterone (DHT), letrozole (LET), testosterone, or IL-5. Currently, there are no animal models that are precise enough to completely mirror the features of PCOS [62,63,64,65,68]. While the DHEA-induced PCOS rodent model displayed key features similar to PCOS in women, such as cystic follicles, it only induced minor metabolic changes in the rats. Notably, however, the cystic follicles in the rodent model had a thinner theca cell layer, contrasting with the thickened theca cell layer seen in human PCOS cases [75]. These findings deviate from the typical characteristics of PCOS, as women with PCOS often experience IR and glucose intolerance. The DHT-induced PCOS model is a valuable tool for examining the mechanisms of altered hormonal regulation and ovarian changes and can be used to investigate various aspects of PCOS such as ovarian function, pathophysiology, metabolic disturbances, and treatment options. This model closely mirrors most of the reproductive and metabolic alterations observed in PCOS women, making it a suitable choice for PCOS research [76,77]. With immunoassay-based techniques, testosterone levels in premenopausal women have been found to be about 40 ng/dL (1.4 nmol/L) and DHT levels about 10 ng/dL (0.34 nmol/L) [60], there were also testosterone mouse model which represent most of the PCOS phenome. In LET-induced PCOS models, metabolic dysregulation such as weight gain increased abdominal adiposity, elevated fasting blood glucose and insulin levels, and IR has been observed [78]. However, the pubertal LET model differs from the organizational effects of androgens during prenatal development, which result in permanent changes in the brain, including alterations in gonadotropin releasing hormone (GnRH) neurons and anxiety-like behavior. Despite this, hyperandrogenism, LH hypersecretion, polycystic ovaries, and a lack of corpora lutea are features found in LET-induced PCOS models and even in ER-α (ESR-1) knockout mice [65,79,80,81]. Therefore, the key features observed in LET-induced models of PCOS may be caused by impaired estrogen action, rather than the hyperandrogenic condition.

#### 2.2.2. Primary GCs Isolated from Normal Animals and Treated with Medicine In Vitro

One of the most common methods of primary GC culture isolation is to isolate cells from ovaries in PCOS rodent models. However, several studies collected GCs from healthy animals and treated them with DHT, IL-15, and H_2_O_2_ [46,60,61,66,67,69,70]. Numerous studies have also demonstrated that ovarian GCs in patients with PCOS experience imbalanced OS, suggesting that autophagy of GCs triggered by OS may play a crucial role in the development and onset of PCOS pathology [46,82]. H_2_O_2_ has been extensively employed as an exogenous inducer in many studies to investigate damage mechanisms. By inducing H_2_O_2_ in rat GCs, researchers have established a model for investigating damage mechanisms such as apoptosis and autophagy that result from OS [60]. It is important to note that the antioxidant system in tissue is more intricate than it is at the cellular level. Therefore, further research is necessary to better understand the complexity of the antioxidant system in tissue and its potential implications in the context of OS and for conditions such as PCOS. Low-grade chronic inflammation may play a role in the development of PCOS, with the proinflammatory cytokine IL-15 contributing to the development of chronic inflammation, which leads to obesity-associated metabolic syndrome. Higher levels of IL-15 in follicular fluid were found in PCOS patients and PCOS animal models, although IL-15 levels in the serum did not increase significantly. It appears, therefore, that IL-15 is produced locally or accumulated in ovarian tissue, such as GC cells. Consequently, GC cells cultured with IL-15 could potentially be used to further explore the pathogenesis of PCOS by affecting the inflammation state, steroidogenesis, and GC survival rates [67].

#### 2.2.3. Other Relevant Cell Lines for PCOS In Vitro Studies

Theca cells (TCs) play a key role during follicular growth and atresia and are one of the most important cell types in the follicles. Their functions include synthesizing androgens, promoting the function of GCs, facilitating oocyte development, and providing structural support for the follicle [83]. Studies have shown that IL-18-dependent regulation of proliferation and steroidogenesis in TCs may influence follicle development and result in similar pathologic features to PCOS [71]. Another study investigated the effects of androgen and the mechanism of Col6a5 in excess lipid accumulation and cell hypertrophy in ovarian stromal cells under a DHT-induced hyperandrogenic mouse model and hyperandrogenic cell models. However, it did not investigate mouse liver cells [72].

### 2.3. Mouse Primary Follicle Cultures

Another method for PCOS studies in vitro is follicle isolation and culture. The ovarian follicle is the fundamental functional unit of the ovary, comprising an oocyte, and surrounding GCs and TCs. The process of folliculogenesis is crucial for the generation of competent oocytes that can be fertilized and developed properly. This process involves the regulation of multiple signals, including hormonal regulation, paracrine signals, and bidirectional communication between the oocyte and the GCs [84]. Follicular development progresses from primordial follicles, primary follicles (PMF), secondary follicles (SF), preantral follicles, antral follicles (AF), and eventually to mature follicles [85]. The abnormal morphology of polycystic ovaries is linked to abnormal ovarian follicle development, which includes increased activation of PMF, ostensible accumulation of follicles at the primary stage, greater proliferation of GCs in small preantral follicles, and follicle arrest at the antral stage [86,87,88,89]. This indicates that ovarian impairment in PCOS originates during the earliest phases of follicle development, when gonadotrophin action is not essential, and where local factors such as androgens play a more prominent role in regulating follicle development [90]. Hence, the in vitro follicle culture system provides an opportunity to study folliculogenesis, oocyte maturation, and pathophysiology.

Hyperandrogenism is detected in PCOS patients. Many studies have suggested that both endogenous and exogenous sources of androgen excess may directly contribute to the development of polycystic ovary morphology [91]. Table 3 has summarized the various rodent primary follicle cultures that have been used to investigate the stimulation of follicle growth by androgens [84,90,92,93,94]. Among the strains of rodents provided by major suppliers, C57BL/6 and ICR mice are commonly used in studies related to aging, toxicology, pathology, and physiology [95]. These two rodent strains are not only easy to handle for their size but also have high fertility. They are also commercially available and widely used for various studies including in vitro culturing of ovarian follicles. In order to investigate follicle development, isolated growing follicles need to be collected from the ovaries of three-week-old C57BL/6 mice or six-week-old ICR mice. DHEA is a precursor to potent androgens and estrogens. In vitro, DHEA supplementation inhibits follicle growth and steroid hormone synthesis. However, blocking AR signaling can reverse the inhibition of follicle growth and ovulation by DHEA [84]. By contrast, DHT significantly promoted secondary follicle growth by potentiating FSH action. Administering DHT to SF in mice with low FSH increased FSHR expression and promoted follicular growth by enhancing FSH action [92]. Testosterone also promoted follicle development in rodent models during the early stages (D7) but inhibited follicle growth at a later stage (D7-14) [94].

In general terms, improper androgen production or conversion to estrogen can lead to hyperandrogenic conditions in PCOS cases [96,97]. The dysfunctional synthesis of androgens and estrogens in PCOS mainly originates from ovarian follicle theca and GCs, resulting in thickened theca and thin granulosa cell layers under polycystic conditions [98]. At the preantral stage, various androgens promote the growth of isolated mouse preantral follicles [99,100,101]. Androgen excess altered the growth patterns of the mice and this hyperandrogenic environment may have contributed to the antral follicle arrest observed in polycystic ovaries [90]. Therefore, rodent primary follicle cultures serve as a good platform to investigate how a hyperandrogenic environment can directly and permanently affect follicular development and ovarian function. 

## 3. Mechanisms and Signaling Pathways Identified from In Vitro Models of PCOS

PCOS is a complex endocrine disease that involves endocrine disorders, androgen excess, infertility, IR, obesity, and glycometabolism disorder [102]. The PI3K-AKT signaling pathway was previously thought to be related to proliferation, apoptosis, and transfer in many diseases, including cancer. However, recent PCOS studies have shown that changes occur in the PI3K-AKT signaling pathway in response to IR, androgen excess, obesity, and follicular development [103]. In addition, emerging evidence indicates that the transforming growth factor beta (TGFβ) signaling pathway and NF-κB signaling pathway also play key roles in regulating the physiological conditions of PCOS [40]. Table 4 summarizes the most important molecular mechanisms involved in PCOS pathogenesis, providing a new perspective on the pathogenesis of PCOS and GC dysfunction.

### 3.1. Proliferation and Apoptosis

GCs are necessary for follicular development and oocyte ovulation as they provide nutrients and growth regulators. Therefore, decreases in GC proliferation and increases in GC apoptosis appear to be significantly involved in the pathogenesis of PCOS [104,105]. The PI3K/AKT/mTOR pathway is an important signaling pathway for cell cycle regulation and cell proliferation [106]. Several studies have indicated that the PI3K/AKT/mTOR pathway plays a role in regulating the proliferation or apoptosis of GCs in follicles (see Table 4) [36,70,107,108,109,110]. In addition, regulators involved in the PI3K/AKT/mTOR pathway, such as ribosomal protein S6 kinase 1 (S6K1), IGF-1, TLR4, and PTEN, may also be associated with the apoptosis/proliferation ratio in GCs of PCOS [36,61,107,109]. The TGFβ signaling pathway is another common pathway involved in multiple cellular processes such as including proliferation, migration, adhesion, and apoptosis, and its dysregulation has been associated with pathological conditions such as PCOS [37,111,112]. Dysregulated miRNAs, including miR-33b, miR-142, miR-125b, miR-203a, and miR-423, have been identified in GCs from PCOS patients and may be involved in regulating the TGFβ-signaling pathway, promoting cell proliferation, and repressing apoptosis [37,40,41]. Furthermore, expression of components in the hedgehog (Hh) signaling pathway has been observed in the GCs of PCOS patients as well as in ovarian tissues in mice [49,113]. Inhibiting the Hh signaling pathway has been shown to decrease GC apoptosis in PCOS, indicating that aberrant activation of the Hh signaling pathway is related to abnormal follicular development in PCOS patients [49]. Recent evidence from fish models suggests that androgens can trigger apoptosis in granulosa/theca (G/T) cells [73,114]. Testosterone activates the membrane receptor ZIP9, which induces G/T cell apoptosis through a mechanism involving Gsα subunit signaling and extracellular signal-regulated kinase (ERK) pathway activation. This mechanism coordinates androgen-induced G protein signaling pathways with zinc signaling to facilitate apoptosis, a crucial cellular function, in vertebrate cells [73].
cells-12-02189-t004_Table 4Table 4Pathological mechanisms and signaling pathways associated with PCOS in various in vitro models.Functional OutcomeAffected PathwayCells/FolliclesSource/TreatmentRef.Proliferation/ApoptosisAKT/mTOR Human primary GCsPCOS patients[107]Proliferation/ApoptosisPI3K/AKT/mTORRat primary GCsND[108]Proliferation/ApoptosisPI3K/AKT/mTORKGNND[109]Proliferation/ApoptosisPI3K/AKT/mTORKGNDHEA[47,115]Proliferation/Apoptosisp53/TGF βHuman primary GC/KGNPCOS patients[40,41]ProliferationTGF βHuman primary GCsPCOS patients[37]ProliferationPI3K/AKTHuman primary GC/KGNmiR-let-7d-3p OE[36]ProliferationPPARγ/PTEN/p-AKTRat primary GCsDHEA/DHT/FSH[61]ApoptosisHedgehog signalingHuman primary GCsPCOS patients[49]Apoptosis/AutophagyIGF1/p-AKT/BimELPorcine primary GCsFSH/PDTC[70]ApoptosisGsα/MAPKfish primary G/T cellsTestosterone[73]AutophagyPI3K/AKT/mTORCOV434ND[116]AutophagyPI3K/AKT/mTORRat primary GCsND[60,64]AutophagyPI3K/AKT/mTORKGNDHEA [47,115]Autophagyp53/AMPKRat primary GCsTP[60,64]AutophagyNF-κB/JNK Porcine primary GCsFSH/PDTC[69]Insulin ResistanceIRS1/PI3K/AKTRat primary GCsDHEA-treated[62]Insulin ResistanceIGF-1/PI3K/AKT/Bax/Bcl-2Rat primary GCsHFD/Letrozole[65]Insulin ResistancePTEN/AKT/TLR2/TLR4/NF-κBHuman primary GCsPCOS patients[42,43]Mitochondrial damageGlycolysis defectionInsulin receptor/PI3K/AKT/ERK KGNSIRT3 KD/Insulin[44]MitophagyPINK1/ParkinHuman primary GC/KGNDHT[48]Mitochondrial dysfunctionPDK1/AKTKGNDHT[110]Oxidative stressPI3K/AKT/mTORRat primary GCsH_2_O_2_[60]Oxidative stressAMPK/AKT/Nrf2Human primary GCsPCOS patients[117]Oxidative stressKeap1/NrF2Human primary GC/Rat primary GCsVitamin K3[46]Estrogen synthesisWNT2/FZD3/β-cateninCOV434FSH[118]Follicular developmentPKA pathwayMice folliclesFSH[119]Abbreviations: mTOR, mammalian target of rapamycin; OE, over expression; PPARγ, peroxisome proliferator-activated receptor gamma; BimEL, bcl-2-interacting mediator of cell death-extra long; AMPK, AMP-activated protein kinase; JNK, Jun amino-terminal kinase; Bax, Bcl-2 associated X; Bcl-2, B-cell lymphoma 2; HFD, high-fat diet; TLR2, toll-like receptor 2; KD, knock down; PINK1, PTEN-induced kinase 1; PDK1, phosphoinositide dependent proteinkinase 1; WNT2, Wnt family member 2; FZD3, frizzled 3; PKA, protein kinase A.


### 3.2. Autophagy

Distinct from apoptosis, autophagy is a cellular process that induces cell death and can facilitate the pathological progression of various diseases [120]. Increasing numbers of studies have revealed that obvious autophagy occurs in GCs in both humans and rats with PCOS and may be one of its primary causes [121]. The PI3K/AKT/mTOR signaling pathway is a classic pathway involved in the regulation of autophagy, and many studies have shown that the activation of the PI3K/AKT/mTOR signaling can inhibit autophagy in both human and rat PCOS models [47,60,116]. The tumor suppressor p53 is another modulator of autophagy in regulating cell death and survival [122]. One study used testosterone propionate to induce a PCOS cell model in vitro by activating autophagy and also showed that autophagy of GCs is inhibited by mediating the p53/AMPK signaling pathway [64]. FSH has been shown to regulate autophagy in rat GCs by activating NF-κB via PI3K/AKT/mTOR signaling [123,124]. By contrast, in porcine GCs, FSH inhibits NF-κB and then promotes autophagy via the JNK pathway, which provides new insights into the crosstalk between different signaling pathways during follicle development [69].

### 3.3. Insulin Resistance

Another key characteristic of PCOS is IR. Several studies have shown that IR regulates multiple mediators and pathways involved in the pathogenesis and development of PCOS [125]. Table 4 shows that insulin primarily regulates the PI3K/AKT signaling to mediate its metabolic regulation effect in both human and rat primary GCs [43,62,65]. Insulin sensitizers, such as humanin analog (HNG) and metformin, suppress IR by regulating the PI3K/AKT signaling pathway [62]. Furthermore, *Cangfudaotan* decoction, a Chinese medicine used to treat gynecological diseases, particularly PCOS, regulates the IGF-1-PI3K/AKT-Bax/Bcl-2 pathway to alleviate IR, improve follicular development, and inhibit apoptosis [65]. Additionally, SAA1 overproduction may contribute to IR development in GCs in PCOS patients by stimulating the TLR2/TLR 4 and NF-κB pathways [42].

### 3.4. Oxidative Stress and Mitochondrial Dysfunction

Numerous studies have increasingly linked low-grade systemic and ovarian chronic inflammation, as well as OS, to PCOS and its associated endocrinological dysfunction [46,126,127]. This indicates that imbalanced OS found in GCs may be an important factor in the development of PCOS pathology. Several studies have reported that OS contributes to the pathophysiology of PCOS through various signaling pathways (see Table 4). In a rat PCOS model, increasing OS-induced GCs autophagy through the PI3K/AKT/mTOR pathway can be ameliorated by treatment with metformin [60], while the increased production of reactive oxygen species (ROS), which causes OS, induced apoptosis in granulosa-lutein cells via activation of the AMPK/AKT/Nrf2 signaling pathway [117]. The Keap1/Nrf2 pathway has recently been identified as the pivotal pathway regulating OS. Nrf2 is a key molecule that becomes activated in response to OS and is usually sequestered by cytoplasmic Keap1 and targeted for proteasomal degradation under basal conditions. Humanin downregulation has been observed in ovarian GCs of PCOS patients and is associated with oxidative imbalance through modulation of the Keap1/Nrf2 signaling pathway [46,128]. Recent studies have indicated that mitochondrial injury in GCs, which is associated with PCOS pathogenesis, is linked to excessive OS [115,129,130]. The PI3K/AKT signaling pathway is one of the major pathways that help stabilize mitochondrial membrane potential and prevent mitochondrial membrane defects, thereby maintaining their primary biological functions. Zheng et al. demonstrated that GCs in PCOS patients contained damaged mitochondrial membranes, and that melatonin might activate PDK1/AKT by promoting SIRT1 expression to repair the damage [110]. By contrast, Yi et al. found that decreased SIRT1 expression in the GCs of PCOS might have caused excessive mitophagy and mitochondrial injury. In addition, melatonin was found to protect against mitochondrial injury in GCs of PCOS by enhancing SIRT1 expression to inhibit excessive PINK1/Parkin-mediated mitophagy [48].

### 3.5. Other Physiological Characteristics and Signaling Pathway

FSH enhances the differentiation capabilities of GCs, including their ability to produce E2 and facilitate preantral follicle growth [131]. Recent research has suggested that Wingless-type MMTV integration site family members (WNTs), working in conjunction with their frizzled (FZD) receptors, contribute to regulating normal folliculogenesis, luteogenesis, and ovarian steroidogenesis. In the cumulus cells (CCs) of patients with PCOS, FZD3 expression was found to be significantly upregulated, which, when coupled with the activation of the WNT2/β-Catenin pathway, was strongly linked to IR and estrogen deficiency. Thus, excessive FZD3 expression in CCs may act as an impediment to steroidogenic activation, which is normally overcome by FSH stimulation [118]. Numerous studies have also provided evidence that FSH can impact the growth of preantral follicles, through in vivo experiments and in vitro culture of ovarian tissue explants or isolated, multilayered preantral follicles. Moreover, FSH-induced growth was found to be suppressed by a PKA inhibitor, indicating that the PKA pathway is involved in FSH-induced follicle growth [119].

## 4. MicroRNA Expression and TARGET Genes in In Vitro Models of PCOS

MicroRNAs (miRNAs) are small, non-coding RNA molecules consisting of approximately 20–22 nucleotides, which negatively regulate target gene expression at the post-transcriptional level by imperfectly base pairing with the 3’-untranslated region (UTR) of target mRNAs. Numerous studies have demonstrated the impact of miRNAs on various biological processes, including development, differentiation, cell proliferation, apoptosis, metabolism, inflammatory responses, and various diseases [132]. miRNA dysregulation is linked to various pathophysiological processes, including PCOS, with increasing evidence suggesting that abnormal expression of miRNAs in GCs plays a crucial role in the onset and progression of PCOS [133,134,135].

Moreover, PCOS patients exhibit distinct patterns of miRNA expression during ovarian steroidogenesis. In women with PCOS, numerous miRNAs are modified in their serum, granulosa-lutein cells (GLCs), and follicular fluids, regulating key processes such as follicular development and maturation, insulin signaling, glucose and lipid metabolism, and steroid hormone synthesis [136,137,138]. A better understanding of how genetic regulation and environmental factors interact to produce varying miRNA expressions may provide valuable insights into the development of PCOS. miRNAs also have the potential to serve as noninvasive biomarkers for PCOS diagnosis and classification. However, the current understanding of the relationship between miRNAs and PCOS development is limited, mainly because a single miRNA can target numerous mRNA molecules and vice versa. Further functional studies exploring the connection between miRNAs and PCOS will be necessary [139].

According to evidence, isolated CCs from PCOS patients exhibited differential expression of several miRNAs compared with controls [140]. Several miRNAs have been shown to increase significantly in GCs from PCOS patients or cell lines. For example, in rat models of PCOS, and in the ovarian GCs of PCOS patients, miR-194 expression was found to be significantly upregulated. Suppressing miR-194 promoted the growth and proliferation of KGN cells, while its over-expression induced cell apoptosis [141]. The upregulation of miR-3188 levels in PCOS patients may enhance cell viability and progression of the cell cycle while suppressing cell apoptosis, an effect that can be achieved through the downregulation of KCNA5 (potassium voltage-gated channel subfamily A member 5) [142]. The expression of miR-21 and toll-like receptor 8 (TLR8) was significantly elevated in granulosa cells of PCOS patients when compared with normal GCs. miR-21 promotes the translation of TLR8 mRNA, leading to increased secretion of IFN-γ, TNF-α, and IL-12, which suggests that miR-21 and TLR8 are involved in PCOS-related inflammation [143]. In PCOS patients, miR-186 and miR-135a were found to be overexpressed. Song et al. identified ESR2 as a direct target of both miR-186 and miR-135a in GCs, establishing a link between dysregulated miRNAs and GC dysfunction in PCOS patients [39] (Table 5).

Several miRNAs were oppositely expressed in GC cells. miR-451a is downregulated in KGN cells, and its upregulation may inhibit GC proliferation, which could be a contributing factor to the development of abnormal follicles in PCOS patients. Additionally, miR-451a regulates the proliferation and apoptosis of ovarian GCs by targeting ATF2 [144]. Both PCOS patients and KGN cells showed downregulation in miR-206 expression. This miRNA targets CCND2, which functions as a negative regulator of miR-206. The regulation of cell viability and apoptosis of ovarian GCs by miR-206 suggests that it may play a critical role in PCOS pathogenesis [145]. High insulin concentrations were found to decrease miR-19b expression levels, promote cell proliferation, and increase IGF-1 levels. Moreover, both PCOS ovary tissues and KGN cells exhibited significantly lower expression levels of miR-19b. Additionally, miR-19b directly targeted IGF-1 and functioned as a negative regulator of its expression. The over-expression of IGF-1 was found to promote cell proliferation [146]. miR-323-3p downregulation was observed in human GCs of women with PCOS, while the inhibition of miR-323-3p levels upregulated steroidogenesis and promoted apoptosis in KGN cells. This miRNA was found to inhibit steroidogenesis and GCs apoptosis by targeting IGF-1, suggesting it may play a role in the development of PCOS [147]. The expression levels of miR-21 were found to be downregulated in the ovarian tissue of PCOS patients and KGN cells. Over-expression of miR-21 inhibited the proliferation of KGN cells and induced apoptosis. The miR-21/SNHG7 axis was shown to play a role in regulating GCs proliferation and apoptosis [148]. The downregulation of miR-320a expression in primary GCs from PCOS patients is associated with estrogen deficiency. IGF1 plays a role in regulating miR-320a expression in GCs. Through direct targeting of the 3’UTR of the osteogenic transcription factor RUNX2, miR-320a potentiates steroidogenesis in GCs by modulating the expression of CYP11A1 and CYP19A1. This suggests that miR-320a may play a critical role in the pathogenesis of PCOS [150]. miR-29a downregulation was observed in PCOS patients and was found to be correlated with an increase in the antral follicle count. The over-expression of miR-29a in KGN and COV434 cells led to the inhibition of cell proliferation, arrested cell cycle progression, and reduced aromatase expression and estradiol production. These findings suggest that miR-29a plays a critical role in GC proliferation and steroidogenesis and could provide new insights into the pathogenesis of PCOS [149].

In light of these findings, miRNAs appear to have the potential to serve as clinical biomarkers for diagnosing PCOS and as therapeutic targets for treating the condition. It may also partly explain the heterogeneity observed in PCOS women. However, despite recent progress, research surrounding miRNAs as potential diagnostic tools is still in the early stages. Additionally, while there is potential for miRNA-based therapeutics, they are yet to be developed. As such, developing commercially available miRNA-based diagnostics and therapeutic tools remains a long-term goal [152,153].

## 5. Pharmacological Approaches in In Vitro Models of PCOS

PCOS has a complex pathophysiology, with an etiology involving multiple factors such as genetics, diet, environment, or even social psychology [47,154]. Current treatments for PCOS tend to focus on symptom relief instead of addressing or curing the root cause. The most commonly used medications and compounds for PCOS therapy are summarized in Table 6. Metformin is a drug usually used to treat patients with PCOS and can improve hyperandrogenism and induce ovulation [60]. In addition to the drugs, there are currently many studies investigating the apoptosis or cell viability of GCs using different nutritional supplements, which could be a new therapeutic modality for PCOS treatment. Another category, Traditional Chinese medicine (TCM) and Herbal Compounds, has its role in PCOS treatment. TCM, a notable component of complementary and alternative medicine, has been used to manage PCOS for centuries. TCM prescriptions comprise multiple active compounds that have been shown to alleviate clinical symptoms and abnormal laboratory data of PCOS without significant side effects [155]. Formulas such as Cangfudaotan, Gui Zhu Yi Kun, Guizhi Fuling Wan, Xiao Yao San, Bu Shen Tian Jing, Bushen Huatan, Kunling Wan, and Gengnianchun have been widely used to treat PCOS in human and animal models. These TCM formulas can inhibit GC autophagy, reduce cell apoptosis, promote normal follicular development, regulate hormone aberrations, ameliorate irregular estrous cycles, reverse ovarian aging, and improve pregnancy outcomes [64,65,156]. Some compounds from herbs, such as cryptotanshinone (CRY) and curcumin, can inhibit granulosa cell apoptosis, regulate OS, improve hormone imbalance, reduce inflammation, and correct disturbances in the estrous cycle in cell and animal models [38,157,158,159]. Additionally, plumbagin exhibits an inhibitory effect on the proliferation and viability of various cancer cell lines and induces a significant concentration-dependent inhibition of rat ovarian GC proliferation. Moreover, a study reported that plumbagin reduce the pyroptosis of GC in PCOS mouse by inhibiting the Wilms tumor 1-associated protein-mediated N6-methylation of C-terminal caspase recruitment domain mRNA. Those study supports the profound potential of plumbagin in PCOS treatment [108,160,161]. All these data show that herbal medicine is an acceptable alternative therapy for PCOS patients. The current therapeutic mechanisms of pharmacological agents for PCOS, as mentioned in Section 4, are highly diverse. Therefore, this section focuses on in vitro studies to better understand their effects.

### 5.1. Small Molecular Drugs

#### Metformin

Metformin is one of the most widely used insulin-sensitizing drugs in the treatment of PCOS. It has also been shown to reduce hyperandrogenism and induce ovulation [166]. However, the use of metformin as an adjunct is limited and only favorable in the treatment of PCOS women who are resistant to CC alone [167,168]. In a rat model, metformin was shown to improve PCOS by decreasing excessive autophagy in primary culture GCs. Metformin also reduces the levels of OS and autophagy in H_2_O_2_-induced GCs through regulating the PI3K/AKT/mTOR signaling pathway [60]. Organic cation transporters (OCTs) OCT1, OCT2, and OCT3 control metformin uptake into primary culture GCs of rats, where metformin can decrease vascular endothelial growth factor (VEGF) and increase pAMPK levels. When OCTs were inhibited, these effects were reversed. Metformin has also been shown to act directly on ovarian cells by regulating cell metabolism and VEGF expression [169].

### 5.2. Nutritional Supplement

#### 5.2.1. Melatonin

Secreted by the pineal gland, melatonin is a neuroendocrine hormone that plays a crucial role in regulating the reproductive functions of mammals [170]. It is a therapeutic agent that regulates autophagy and also has the potential to suppress autophagy and apoptosis in PCOS [47,171]. Additionally, studies have shown that melatonin can significantly reduce androgen levels and increase FSH levels in PCOS patients. Moreover, it has a positive effect on oocyte quality in cases of hypoestrogenia and hyperandrogenia [45,172].

Melatonin is not only used in PCOS treatment, it is also a mitochondria-targeted antioxidant and might repair mitochondrial damage [43,48,110,173]. In both GCs from PCOS patients and DHT-treated KGN cells, melatonin was found to increase the expression of SIRT1 and decrease the expression of PINK1/Parkin at the protein level, which improves mitochondrial dysfunction [48]. Melatonin decreases mitochondrial permeability transition pore (mPTP) opening and increases the JC-1 aggregate/monomer ratio, indicating that regulating mPTP can enhance mitochondrial membrane potential. It reduces the levels of cytochrome C and Bax both in vivo and in vitro while increasing the phosphorylation of PDK1 and AKT. This activation of the PDK1/AKT pathway is crucial in improving mitochondrial membrane function. Additionally, melatonin treatment increases the expression of SIRT1, while knocking down SIRT1 mRNA inhibits the protective effect of melatonin on the activation of PDK1/AKT [110].

Melatonin has been shown to have multiple effects on luteinized GCs of PCOS patients. Firstly, it upregulates the expression of CYP19A1 via the ERK pathway, which accelerates the conversion of androgen to 17β-estradiol. Secondly, it reduces the levels of inducible nitric oxide (NO) synthetase and NO in GCs. Thirdly, it increases the level of transcripts encoding Nrf2 and its downstream target HO1, which results in anti-inflammatory and antioxidant effects [45].

Melatonin has been shown to improve glucose uptake and insulin signaling in both GCs from PCOS patients and SVOG cells that were treated with palmitic acid. In these cells, melatonin increases IRS-1 and GLUT4 expression while decreasing p-IRS-1 (Ser307) expression. Palmitic acid has been shown to inhibit PI3K and AKT phosphorylation, but melatonin increases the levels of p-PI3K and p-AKT while decreasing IR via the PI3K/AKT signaling pathway in GCs and palmitic acid-induced SVOG cells [43].

#### 5.2.2. Humanin

Humanin, a peptide derived from mitochondria, is involved in metabolic processes such as diabetes and PCOS. Wang et al. showed that local ovarian expression of humanin was downregulated in PCOS patients with IR compared with its expression in PCOS patients without IR. They also showed that exogenous humanin supplementation could improve body weight gain, ovarian morphological abnormalities, endocrinological disorders, and ovarian and systemic OS in PCOS rat models induced by DHEA [46].

Activating the Keap1/Nrf2 signaling pathway in response to OS is one mechanism by which humanin exerts its protective effects on ovarian GCs in patients with PCOS. By modulating this pathway, humanin can alleviate OS in GCs [46]. A study conducted that supplementation with a humanin analog could reduce elevated levels of fasting plasma glucose and fasting insulin in PCOS rat models induced by DHEA [62]. Moreover, exogenous humanin supplementation has been shown to improve the metabolic profile of PCOS rats by targeting the IRS1/PI3K/AKT insulin signaling pathway, and it decreased the phosphorylation of IRS1, PI3K, AKT, and GLUT4 proteins in the primary culture of ovarian GCs [62]. These results suggest that humanin has great potential as a therapeutic drug for patients with PCOS.

#### 5.2.3. Vitamin D3

In PCOS patients, Vitamin D deficiency can lead to excessive androgen secretion, IR, and disrupted follicular growth. However, Vitamin D3 can increase the number of preantral and antral follicles in a DHEA-induced rat PCOS model compared with a healthy group [68]. Vitamin D3 was shown to improve mitochondrial biogenesis, membrane integrity, and mtDNA copy number in primary culture GCs of PCOS mice induced with DHEA, potentially enhancing follicular development and oocyte quality [68].

#### 5.2.4. Sulforaphane

Sulforaphane has anti-tumor, immunoregulatory, and antioxidative effects, and can protect against OS by lowering the level of intracellular ROS and apoptosis in the GLCs of PCOS patients [117]. Sulforaphane activates the AMPK/AKT/Nrf2 signaling pathway, providing protection against OS in the GLCs of PCOS patients [117].

### 5.3. Traditional Chinese Medicine

An in vitro Gui Zhu Yi Kun formula study showed that primary culture GCs treated with Gui Zhu Yi Kun formula can increase in mTOR, phosphorylated mTOR, and AMPKα expression levels, and reduce p53 and sestrin2 expression levels. This finding implies that a decrease in ovarian primary culture GCs in rats with PCOS may be related to autophagy [64]. The Cangfudaotan decoction has been shown to improve IR and diminish ovary morphological damage, normalize abnormal serum hormone levels, and inhibit inflammatory cytokines in a PCOS rat model [65]. In the primary culture GCs studies, Cangfudaotan can improve cell viability and inhibit cell apoptosis, which is associated with the regulation of IGF-1-PI3K/AKT-Bax/Bcl-2 pathway-mediated gene expression [65]. Guizhi Fuling Wan has fewer atretic and cystic follicles, and more mature follicles and corpus lutea, as well as lower serum T, LH, LH/FSH ratios, HOMA-IR, and FINS levels in the PCOS rat model [164].

Other studies have demonstrated that Guizhi Fuling Wan can inhibit autophagy in primary culture GCs and promote follicular development to attenuate ovulation disorder in PCOS-IR, which is associated with the activation of the PI3K/AKT/mTOR signaling pathway [156]. Xiao Yao San can reduce the level of autophagy in GCs caused by noradrenaline and can also mitigate the autophagy of primary culture GCs through the AKT/mTOR/S6K1 pathway [162]. Additionally, Xiao Yao San has been shown to inhibit the apoptosis of primary culture GCs in the AF and the autophagy of GCs in the antral and cystic follicles in a chronic, unpredictable mild stress-induced PCOS rat model [162]. Bu Shen Tian Jing can ameliorate glucose tolerance, the estrous cycle, and ovarian morphology in the PCOS rat model, while Bu Shen Tian Jing relieves OS and increases SIRT3 expression of ovarian GCs [163]. In the above study on the KGN cell model, Bu Shen Tian Jing was shown to reverse palmitate-induced impaired SIRT3 expression and glucose uptake and decrease palmitate-induced mitochondrial ROS production mediated by SIRT3 [163]. Both Bushen Huatan and Kunling Wan can improve pregnancy outcomes, improve the viability of primary culture GCs and decrease the apoptosis of primary culture GCs in a PCOS rat model [164]. The study also showed that Bushen Huatan granules inhibit apoptosis by attenuating the dysfunctional mitochondrial of primary culture GC, while Kunling Wan can relieve endoplasmic reticulum stress [164]. Gengnianchun can decrease IR, reduce damage to the ovarian reserve, and reduce aging-related mRNA and protein levels such as p53, p16, and p21 in a mice PCOS model [165]. In a long-term insulin-treated KGN cells model, which results in senescence, differently expressed genes are mainly enriched to the ERK1 and ERK2 cascade pathway, which is an integral component of the plasma membrane and signals a level of receptor binding [165].

### 5.4. Herbal Compounds

In vitro models of compounds from herb plants, such as CRY, have been shown to reduce OS, decrease high mobility group box 1 (HMGB1) and Bax expression of primary culture GCs induced by ischemia-reperfusion through inhibiting ferroptosis, reduce GPX4 expression, and activate NF-κB via the MAKP signaling pathway [157]. Another study of CRY demonstrates that CTBP1-AS is highly expressed in PCOS patients. CTBP1-AS interacts with EZH2 and EED in primary culture GCs and balances the proliferation and apoptosis of primary culture GCs in PCOS patients. CRY has been shown to reduce the level of CTBP1-AS in a KGN cell model [38] and has also been shown to attenuate increases in body weight, ovarian quotiety, Lee’s index, and body mass index in a PCOS rat model [158]. CRY also reduces the proliferation of primary culture GCs and modulates TNF-α, TLR4, NF-κB/p65, and HMGB1 expression in vitro [158].

Curcumin can reverse the phenotype of PCOS model rats such as irregular estrus cycle, the increase in body and ovarian weight, the elevation of serum T and LH, and the decrease in FSH [159]. Curcumin also attenuates ER stress in primary culture GCs by activating the PI3K/AKT pathway and protects against the apoptosis of primary culture GCs, perhaps by activating the PI3K/AKT pathway and inhibiting IRE1α-XBP1 levels [159]. Plumbagin can deactivate the PI3K/AKT/mTOR pathway in the primary GCs of the PCOS rat model, which results in the apoptosis and inhibition of proliferation of the primary GCs.

In summary, compounds from herb plants have multiple regulatory mechanisms in GCs such as the MAKP signaling pathway, the PI3K/AKT pathway, HMGB1, TNF-α, TLR4, and NF-κB/p65, resulting in a more regular estrous cycle and normal serum testosterone, LH, and FSH levels in animal models.

## 6. Conclusions

Experimental models have been extensively used to gain a better understanding of PCOS. However, merely treating the symptoms is not enough; suitable models must be utilized to develop a cure for PCOS. While clinical observations have supported the role of androgen actions in PCOS, there is currently a lack of conclusive evidence. Recent experimental studies have provided substantial evidence supporting the significance of direct AR-mediated androgen actions in the development and progression of PCOS. However, because of the ethical limitations surrounding human experimentation, animal models using rodents, sheep, and non-human primates have had to be developed. Nevertheless, in vitro models of PCOS offer a unique opportunity for mechanistic experiments to be conducted and be reflected pathological features of PCOS (see Table 7), and provide insights into the pathogenesis of PCOS (Figure 2). In summary, by combining clinical observations with targeted in vitro models, we can have a better understanding of the mechanistic pathogenesis of PCOS. This approach offers the possibility of identifying target sites and key pathways involved in PCOS, which may, in turn, lead to the development of novel, evidence-based therapeutic treatment for PCOS.

## Figures and Tables

**Figure 1 cells-12-02189-f001:**
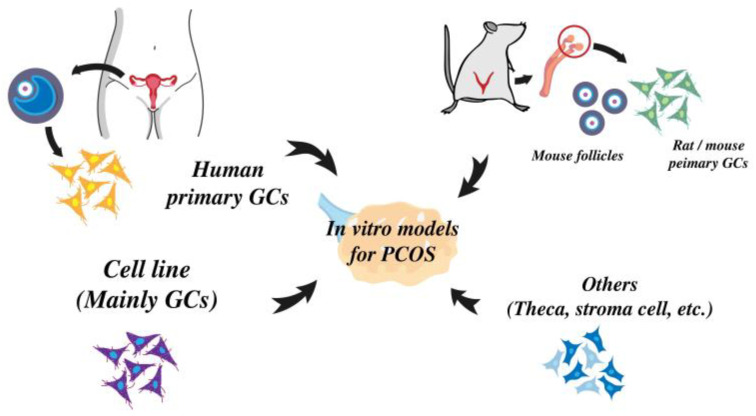
The common in-vitro models used in PCOS studies are outlined. The primary GCs from humans and rodents, cell lines, and others related to the reproductive system or endocrine system are utilized in PCOS research.

**Figure 2 cells-12-02189-f002:**
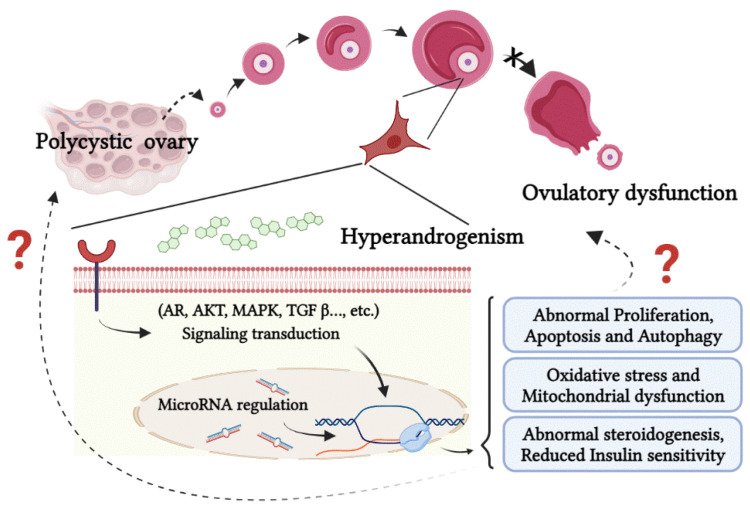
PCOS is a disorder characterized by polycystic ovaries, hyperandrogenism, and ovulatory dysfunction. The common concepts pertaining to PCOS in in-vitro studies are shown. In-vitro models can be used to study signal transduction, microRNA regulation, and subsequent biological processes.

**Table 2 cells-12-02189-t002:** Primary cells from the ovary and other relevant cells for in vitro PCOS studies.

Human	Manipulation/Treatment	Mainly Description	Target	Ref.
Primary granulosa cells	from women underwent IVF	Proliferation	TLR4	[36]
			TGFBR1, SMAD7	[37]
		Proliferation, apoptosis	CTBP1-AS	[38]
			ESR2	[39]
			TGFBR1, TGFBR2	[40]
			MALAT1, MDM2	[41]
		Insulin resistance	SAA1	[42]
			IRS-1, GLUT4	[43]
		Mitochondrial damage Glycolysis defection	SIRT3	[44]
		Androgen production Antioxidative damage	CYP19A1, HO1	[45]
		Oxidative stress	Keap1, Nrf2	[46]
		Autophagy	Beclin-1, light chain 3	[47]
		Mitophagy	SIRT1	[48]
		Ovulation	FHL2, AR	[13]
		RNA sequencing	TNF-α	[49]
NCI-H295R (Adrenocortical carcinoma)	Decanoic acid	Androgen production	3β-HSDII	[54]
	Forskolin	Androgen production	CYP17A1, 3β-HSDII	[55]
	ND	Androgen production	3β-HSDII	[56]
	BMP2	Steroidogenesis, androgen production	BMP2	[57]
NCI-H295R (Adrenocortical carcinoma)	ND	Steroidogenesis, androgen production	CYP17, CYP21	[58]
NCI-H295R (Adrenocortical carcinoma)	Palmitate, H_2_O_2_, HNE	Steroidogenesis, androgen production	P450c17	[59]
**Animal**				
Rat primary granulosa cells	H_2_O_2_ (in vitro)	Oxidative stress, autophagy	Beclin-1, light chain 3	[60]
	DHT and FSH (in vitro)	Proliferation	PTEN	[61]
	DHEA (in vivo)	Mitochondrial abnormality, insulin resistance	NDUFB8, ATP5j, IRS-1, GLUT4	[62,63]
	TP (in vivo)	Autophagy	Beclin-1, light chain 3	[64]
	Vitamin K3 (in vitro)	Oxidative stress	Keap1, Nrf2	[46]
	Letrozole (in vivo)	Insulin resistance	IGF-1, PI3K, AKT	[65]
Mouse primary granulosa cells	DHT (in vitro)	Autophagy	Beclin-1, light chain 3	[66]
	IL-15 (in vitro)	Proliferation, apoptosis, steroidogenesis, inflammation	CYP17A1, Ifng, IL-1b	[67]
	DHEA (in vivo)	Mitochondrial biogenesis	mtDNA	[68]
Porcine primary granulosa cells	FSH and PDTC (in vitro)	Autophagy, apoptosis	NF-κB, IGF-1	[69,70]
Bovine primary theca cells	IL-18 (in vitro)	Proliferation, steroidogenesis	CYP11A1, CYP17A1	[71]
Mouse primary stroma cells	DHT and LH (in vitro)	Lipid metabolism disorder, steroidogenesis, hypertrophy	Col6a5	[72]
Fish primary granulosa/theca cells	Testosterone (in vitro)	Apoptosis	MAPK	[73]
3T3-L1 (Mouse preadipocyte)	Testosterone, LPS	Inflammation	IL-6 MCP-1	[74]
	DHT	Lipid metabolism disorder	Col6a5	[72]
NCTC1469 (Mouse liver cells	DHT	Lipid metabolism disorder	Col6a5	[72]

Abbreviations: TLR4, toll-like receptor 4; TGFBR, transforming growth factor beta receptor; SMAD7, mothers against decapentaplegic homolog 7; CTBP1-AS, C-terminal binding protein 1-antisense RNA; ESR2, estrogen receptor 2; MALAT1, metastasis associated lung adenocarcinoma transcript 1; MDM2, mouse double minute 2; SAA1, serum amyloid A1; IRS-1, insulin receptor substrate 1; GLUT4, glucose transporter type 4; SIRT, sirtuin; CYP19A1, cytochrome p450 family 19 subfamily A member 1; HO1, heme oxygenase 1; Keap1, kelch-like ECH-associated protein 1; Nrf2, nuclear factor like 2; FHL2, four and a half LIM domains 2; AR, androgen receptor; TNF-α, tumor necrosis factor- alpha; CYP17A1, cytochrome p450 family 17 subfamily A member 1; BMP2, bone morphogenetic protein 2; CYP21, cytochrome p450 family 21; HNE, hexanitroethane; PTEN, phosphatase and tensin homolog; NDUFB8, NADH: ubiquinone oxidoreductase subunit B8; ATP5j, ATP synthase peripheral stalk subunit F6; TP, testosterone propionate; IGF-1, insulin like growth factor 1; PI3K, phosphatidylinositol 3-kinase; AKT, protein kinase B; Ifng, interferon gamma; IL, interleukin; mtDNA, mitochondrial DNA; PDTC, pyrrolidine dithiocarbamate; NF-κB, nuclear factor kappa B; CYP11A1, cytochrome p450 family 11 subfamily A member 1; LH, luteinizing hormone; Col6a5, collagen type VI alpha 5 chain; MAPK, mitogen-activated protein kinase; LPS, Lipopolysaccharide; MCP-1, The monocyte chemotactic protein-1.

**Table 3 cells-12-02189-t003:** Characterization of mouse primary follicle cultures from androgen-induced PCOS models.

Mice Species	C57BL/6J	C57BL/6J	ICR	ICR	Kunming
**Sacrificed age**	Day 18 to 21	Week 13 to 16	Week 6	Week 6	Day 14
**follicle stage**	Secondary follicle	Small preantral follicle (100 to 150 μm) Large preantral follicle (151 to 200 μm) Small antral follicle (201 to 250 μm) Large antral follicle (251 to 350 μm) Preovulatory follicle (351 to 450 μm)	Secondary follicle (100 to 160 μm)	Secondary follicle (100 to 160 μm)	Primary follicle
**Culture Period**	6 days	5 days	13 days	13 days	14 days
**Culture medium**	αMEM + 10 mIU/mL rFSH	αMEM + 100 mIU/mL rFSH	αMEM + 33 mIU/mL rFSH or 100 mIU/mL rFSH	αMEM + 33 mIU/mL rFSH	αMEM + 100mIU/mL rFSH and LH
**Culture dish**	96 well dish	4 well dish	48 well dish	96 well dish	10 μL droplet
**Treatment**	10 μM DHEA, 10 μM flutamide	DHT	50, 500, 1250 ng/mL DHT	500 ng/mL DHT 5 ng/mL Pioglitazone	10^−6^, 10^−5^, 10^−4^ M Testosterone
**Reported outcome**	DHEA induces impairment of follicle growth and ovulation	Prolonged exposure to excess DHT leads to aberrant follicle development	DHT supports follicle development during the FSH-dependent preantral stage	Pioglitazone negatively affects follicular growth	Testosterone promotes the follicle development
**Ref.**	[84]	[90]	[92]	[93]	[94]

Abbreviations: αMEM, alpha minimal essential media; rFSH, recombinant FSH.

**Table 5 cells-12-02189-t005:** MicroRNA expression and target genes in in vitro PCOS studies.

MicroRNA	Target	Cells	Cell Response	Ref.
miR-451a	ATF2	KGN	Proliferation ↓, apoptosis ↑	[144]
miR-194	HB-EGF	KGN	Proliferation ↓, apoptosis ↑	[141]
miR-3188	KCNA5	KGN	Proliferation ↑, apoptosis ↓	[142]
miR-206	CCND2	KGN	Proliferation ↓, apoptosis ↑	[145]
miR-let-7d-3p	TLR4	KGN	Proliferation ↓	[36]
miR-19b	IGF-1	KGN	Proliferation ↓	[146]
miR-323-3p	IGF-1	KGN	Steroidogenesis, apoptosis	[147]
miR-21	SNHG7	KGN	Proliferation ↓, apoptosis ↑	[148]
miR-186, miR-135a	ESR2	KGN	Proliferation ↑, apoptosis ↓	[39]
miR-29a	ND	KGN, COV434	Proliferation ↓, Aromatase expression↓, Estradiol biosynthesis ↓	[149]
miR-320a	RUNX2	Human primary GCs	Steroidogenesis	[150]
miR-21	TLR8	Human primary GCs	Proliferation ↑, apoptosis ↓	[143]
miR-33b, miR-142	TGFBR1	Human primary GCs	Proliferation ↑, apoptosis ↓	[37]
miR-423	SMAD7	Human primary GCs	Proliferation ↑, apoptosis ↓	[37]
miR-324-3p	WNT2B	Rat primary GCs	Proliferation ↓, apoptosis ↑	[151]

↓, decrease; ↑, increase. Abbreviations: ATF2, cyclic AMP-dependent transcription factor 2; HB-EGF, Heparin-binding epidermal growth factor-like growth factor; CCND2, cyclin D2; SNHG7, small nucleolar RNA host gene 7; RUNX2, Runt-related transcription factor 2.

**Table 6 cells-12-02189-t006:** Pharmacological agents used to investigate the pathophysiology of PCOS in in vitro models.

Agents for Treatments	Cells	Source/Treatment	Treatment Outcome	Ref.
Metformin	Rat primary GCs	H_2_O_2_	Decreased excessive autophagy in GCs	[60]
Melatonin	Human primary GCs	PCOS patients	Reduced insulin resistance in GCs	[43]
Melatonin	Human primary GCs	PCOS patients	Reduced androgen levels through ERK in GCs	[45]
Melatonin	KGN cell	DHT	Protected against mitochondrial injury in GCs of PCOS	[48]
Melatonin	KGN cell	DHT	Ameliorated mitochondrial membrane damage in GCs of PCOS	[110]
Melatonin	KGN cell	DHEA	Suppressed autophagy and apoptosis	[47]
HNG supplementation	Rat primary GCs	DHEA-induced	Improved local ovarian insulin resistance	[62]
HNG supplementation	COV434	Vitamin K3	Alleviated oxidative stress in GCs of PCOS	[46]
Vitamin D3	Mice Primary GCs	DHEA	Improved mitochondrial biogenesis and membrane integrity	[68]
Sulforaphane	Human primary GCs	PCOS patients	Against oxidative stress by recudcing intracellular ROS and apoptosis levels	[117]
Gui Zhu Yi Kun formula	Rat primary GCs	TP	Inhibited GC autophagy	[64]
Cangfudaotan Decoction	Rat primary GCs	Letrozole	Suppressed insulin resistance and improves follicular development	[65]
Guizhi Fuling Wan	Rat primary GCs	Letrozole	Inhibited GC autophagy and promoted follicular development	[156]
Xiao Yao San	Rat primary GCs	Chronic unpredictable mild stress	Reduced apoptosis and autophagy of GCs	[162]
Bu Shen Tian Jing Formula	KGN	Palmitate	Improving oxidative stress and glucose metabolism	[163]
Bushen Huatan Granules and Kunling Wan	Rat primary GCs	DHEA	Protected endoplasmic reticulum stress.	[164]
Gengnianchun recipe	KGN	Insulin	Inhibited the senescence of GCs	[165]
Cryptotanshinone	Rat primary GCs	hCG and insulin	Inhibited oxidative stress and apoptosis	[157]
Cryptotanshinone	KGN	ND	Inhibited the proliferation and promote the apoptosis	[38]
Cryptotanshinone	Rat primary GCs	hCG and insulin	Attenuated hormone and inflammatory factor level	[158]
Curcumin	Rat primary GCs	DHEA	Inhibited endoplasmic reticulum stress	[159]
Plumbagin	Rat primary GCs	ND	Inhibited proliferation and promotes apoptosis	[108]
Plumbagin	mice primary GCs	DHEA	Reduced the pyroptosis of GCs	[161]

**Table 7 cells-12-02189-t007:** The connection between parameters in in vitro models and pathological features of PCOS.

Parameters	Experimental Measurements	Pathological Features of PCOS	
**Morphology**			
Apoptosis/Autophagy	Flow cytometry, caspase 3 activity assay, TUNEL staining, LC3 Ⅱ/LC3 Ⅰ ratio, follicle survival rate	Follicular atresia, follicular arrest	[38,47,92]
Mitochondria structure	Electron microscopy, mtDNA copy number measurement	Mitochondrial dysfunction, glycolysis defection, impaired oocytes	[68]
Proliferation	Cell proliferation assay, follicle diameter measurement	Disruption of folliculogenesis, follicular arrest	[38,84]
**Biological function**			
Lipid metabolism	Triglyceride and Total Cholesterol measurement, Oil Red O Staining	Dyslipidemia	[72]
Ovulation	hCG-induced ovulation test, RT-PCR for ovulation related gene	Oligo/Anovulation	[84]
Steroidogenesis	ELISA, RT-PCR for steroidogenesis related gene	Steroidogenesis imbalance	[57]
**Chemical/Stress response**			
Androgens	RT-PCR for AR related gene, western blotting	Hyperandrogenism	[13]
Cytokines	RT-PCR for pro-inflammatory cytokine related gene, immunofluorescence, ELISA	Chronic inflammation	[67]
Insulin	IRS-1/GLUT4 RT-PCR, Glucose Consumption Assay western blotting, immunofluorescence	Insulin resistance	[62]
Oxidative stress	MDA, SOD, and GSH detection	Chronic inflammation, glycolysis defection, impaired oocytes, insulin resistance	[72]

Abbreviations: TUNEL, Terminal deoxynucleotidyl transferase dUTP nick end labeling; LC3: Microtubule-associated protein light chain 3; RT-PCR: Reverse transcriptase PCR; ELISA: Enzyme-linked immunosorbent assay; MDA: Malondialdehyde; SOD: Superoxide dismutase; GSH: Reduced glutathione.

## Data Availability

Not applicable.

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
