# Peer review of "Current Advances in Cellular Approaches for Pathophysiology and Treatment of Polycystic Ovary Syndrome"

_cells, 2023, doi:10.3390/cells12172189_

Round 1

Reviewer 1 Report

Summary:

In the review article, Tsai et al., overviewed the literatures in regarding to cellular models for studying PCOS disease. The theme of this article is similar to a recent published paper in Life Science (Life Sciences, 322, 1 June 2023, 121672), however, the current paper is more comprehensive. The current article summarized human and rodent resources of cell tools with genetic profile, cellular characteristics, and corresponding signaling. In general, this work is timely important and will be benefit to this particular field. There’re some minor deficits in the manuscript and needed to be modified before accept for publication.

Critiques:

1.   There’re two important themes for modeling tool. One for reagents used in modeling, the other for the parameters of measurements which represent diseases. The author already described nicely on the reagent, e.g., in vitro cell tools. However, the parameters for measuring varieties of pathological alternation is missed. Although some description sporadically in the texts, it’s important to summarize in tables to enlist measurements vs. pathological feature/therapeutic responses…etc.

2.   The culturing medium for each assay is important information for researchers and biotech industry. For example, some ligands/chemical/peptide/growth factors as supplements in the medium for certain assay usually crucial, experimentally and biologically. If possible, please compare the culture reagents or condition/environment of in vitro experiments. It’ll be benefitial to specialists and industrial developer. Although table 3 descript in that sense, it’s not the scenario of reviewer suggestion. It’s definitely can be improved based on Table 3 current format.

3.   In line 199 (page 7), the separation of single sentence as one paragraph reads odds to the readers. It’s pointless distinguishing this sentence with the following paragraph. Reviewer suggest to merge.

Reviewer 2 Report

An interesting review article highlighting the current knowledge and research in the field of PCOS. This is a well-written article. However, if authors can add one or two schematic diagrams to demonstrate the signaling challenges during ovulation and hormonal action.   

Minor English language editing may require. 
